mathematical modelling/computational biology/health and disease and epidemiology

COVID-19, SARS-CoV-2, ageing, parameter estimation, mathematical model

**Author for correspondence:**
Hana M. Dobrovolny
e-mail: h.dobrovolny@tcu.edu

# Estimation of viral kinetics model parameters in young and aged SARS-CoV-2 infected macaques

Thalia Rodriguez and Hana M. Dobrovolny

Department of Physics and Astronomy, Texas Christian University, Fort Worth, TX, USA

HMD, 0000-0003-3592-6770

The SARS-CoV-2 virus disproportionately causes serious illness and death in older individuals. In order to have the greatest impact in decreasing the human toll caused by the virus, antiviral treatment should be targeted to older patients. For this, we need a better understanding of the differences in viral dynamics between SARS-CoV-2 infection in younger and older adults. In this study, we use previously published averaged viral titre measurements from the nose and throat of SARS-CoV-2 infection in young and aged cynomolgus macaques to parametrize a viral kinetics model. We find that all viral kinetics parameters differ between young and aged macaques in the nasal passages, but that there are fewer differences in parameter estimates from the throat. We further use our parametrized model to study the antiviral treatment of young and aged animals, finding that early antiviral treatment is more likely to lead to a lengthening of the infection in aged animals, but not in young animals.

## 1. Introduction

The novel coronavirus, SARS-CoV-2, has caused widespread illness and death around the world [1]. While it appears that a substantial fraction of people infected with the virus are asymptomatic [2,3], some people develop severe symptoms that lead to respiratory failure [4–6]. One of the biggest risk factors for developing severe disease is age [7,8], with studies suggesting that both the risk of severe disease and mortality increase with age [9,10].

There are several possible reasons for the increased likelihood of severe disease in older adults. There are a number of co-morbidities associated with severe COVID-19 that become more common with age [11,12]. Mucociliary clearance becomes less effective as people age [13,14], resulting in increased likelihood of infection when exposed to respiratory pathogens such as SARS-CoV-2. Ageing also results in chronic, subclinical inflammation (referred to as

'inflammaging') that increases the likelihood of serious disease during infections [15–17]. Additionally, a weakening of the immune response, known as immunosenescence, is known to increase susceptibility to severe illness during viral infections [17–20].

While there are general age-related changes that are thought to affect the severity of SARS-CoV-2 infections, there are studies that have found specific age-related changes in the body's response to SARS-CoV-2. A study comparing SARS-CoV-2 infection in rhesus macaques noted that the immune response to the virus is delayed in aged animals [21]. A study in young and aged Syrian hamsters, showed a similar delayed immune response in the aged animals [22]. There is also a change in the types of immune cells that respond to SARS-CoV-2 in older patients [23]. There is decreased expression of Th1 chemokines, resulting in a reduced T-cell response [24,25], but an increase in circulation of several cytokines [26]. A part of this dysfunctional immune response leads to a more widespread and severe inflammatory response in the elderly [27]. Finally, there are findings that the expression of some cell receptors involved in SARS-CoV-2 replication changes with age [23,28,29].

While many studies have examined the differences in clinical presentation of SARS-CoV-2 infection in young and aged patients [30–32], there has not been an assessment of how viral time courses differ in young and old individuals. Quantitative analysis of viral time courses through parametrization of mathematical models can provide insight into the viral replication cycle. In particular, comparative analysis has led to a better understanding of viral life cycle differences between wild-type and drug-resistant strains of influenza [33,34] by finding a lengthening of the eclipse phase duration in the mutants with mutant fitness determined by the presence or absence of a compensatory trait that increases viral infectiousness. A similar comparative analysis was used to elucidate how the phases of the viral replication cycle of respiratory syncytial virus (RSV) differ in *in vitro* and various *in vivo* systems [35]. Comparison of model parameters also determined that the key dynamical difference between influenza and RSV viruses is differences in the infecting time [36]. Finally, fitting of mathematical models was used to compare viral titres in untreated and remdesivir-treated rhesus macaques infected with SARS-CoV-2 to help determine the effect of remdesivir on viral kinetics [37].

In this manuscript, we fit a mathematical model of viral infection to parametrize averaged viral titre time courses from young and aged cynomolgus macaques. We find differences in the viral titre curves and the underlying dynamical parameters between young and aged animals in the nasal passages, but many of these differences do not extend to the viral titre curves in the throat. Importantly, these differences alter the effectiveness of antivirals in shortening the duration of the infection in aged animals.

# 2. Methods

## 2.1. Experimental data

The data used in this manuscript were taken from Rockx *et al.* [38], who studied pathogenesis of SARS-CoV-2 in cynomolgus macaques. Briefly, two groups of female macaques, four young adults of 4–5 years of age and four older adults of 15–20 years of age, were inoculated with SARS-CoV-2. The virus was introduced through a combined intratracheal and intranasal route with a total initial infectious dose of $10^6$ TCID$_{50}$. Nasal and throat swabs were taken on days 0, 1, 2, 3, 4, 6, 8, 10, 14, 18 and 21 for RT-qPCR analysis. Published data consist of a single averaged viral titre curve for each of the following: young nasal, aged nasal, young throat, aged throat. Data were extracted from Fig. 1A & B of [38] using WebPlotDigitizer (https://automeris.io/WebPlotDigitizer/).

## 2.2. Mathematical model

We use a standard ordinary differential equation (ODE) model of viral infection [39] to determine viral kinetics parameters for the old and young macaques,

$$\left. \begin{aligned} \frac{dT}{dt} &= -\beta TV, \\ \frac{dE}{dt} &= \beta TV - kE, \\ \frac{dI}{dt} &= kE - \delta I \\ \frac{dV}{dt} &= pI - cV. \end{aligned} \right\} \tag{2.1}$$

and

In the model, target cells $T$ are infected by virus $V$ at a rate $\beta$. The cells then transition to the eclipse phase, $E$, where they are infected but not yet producing virus. After a time $1/k$, the cells transition to the infectious phase, $I$, where they are actively producing virus at a rate $p$. The infectious cells die after a time $1/\delta$ and virus is cleared from the system at a rate $c$. In addition to model parameters, we calculate the basic reproduction number for the model, $R_0 = \beta p T_0 / c\delta$, which gives the number of secondary infections arising from a single infected cell. We also calculate the infecting time, $t_{\text{inf}} = \sqrt{2/\beta p T_0}$, which is the time between release of virus in one cell and infection of the next cell (derivation found in [36]).

## 2.3. Data fitting and statistical analysis

In order to reduce the number of free parameters, we fix the initial number of target cells, $T(0) = 1$, meaning we are using units of relative number of cells. We also assume that the infection is initiated with some amount of virus, meaning $E(0) = I(0) = 0$. The initial amount of virus is determined via fitting. We also fix the transition rate from the eclipse phase ($k = 3 \, \text{d}^{-1}$) and the clearance rate ($c = 10 \, \text{d}^{-1}$) as in [40,41]. Best-fit parameters are determined by minimizing the sum of squared residuals (SSR) using the `minimize` function in Python's `scipy` package. The `scipy.minimize` function includes several possible minimization algorithms and we used several different algorithms (Nelder-Mead, truncated Newton, L-BFGS-B and trust-constraint) in an effort to find the best fit. For experimental data at the detection threshold, model estimates only contribute to the SSR if they are above the threshold. Since we have a single viral titre curve for each group, we cannot evaluate the error in estimated parameter values using measurements from multiple animals. Instead, we evaluate parameter distributions using 1000 bootstrap replicates [42,43] to estimate parameter variability.

Statistical comparisons are performed using the Mann–Whitney (Wilcoxon rank-sum) test, which is used to compare two distributions without assuming normality. To avoid over-powered analysis, we use 100 random samples of 10 estimates from each distribution to perform the Mann–Whitney test and report the mean $p$-value. A mean $p$-value of less than 0.01 is considered statistically significant.

# 3. Results

## 3.1. Model fit to data

We fit the ODE model to the experimental data from both the throat and nasal swabs of young and aged macaques. The model fits are shown in figure 1 with parameter estimates given in table 1. Parameter correlation plots and likelihood profiles are included in the electronic supplementary material. Data show that the viral infections last longer in the aged animals. The time of peak is early in the nasal swabs of young animals and in the throat swabs of both young and aged animals. There is a delayed time of peak in the nasal swabs of aged animals. Viral loads also reach higher peaks in the nose than in the throat in both young and aged animals, with the highest viral loads observed in the nasal passages of aged animals.

In terms of model parameters, there appears to be a lower infection rate in both the nasal passages and throats of aged macaques than in young macaques. The death rate of infectious cells is higher in both the nose and throat of young macaques than in aged macaques. The basic reproduction number in young animals is higher than the reproduction numbers found for aged animals; however, the variability in the estimates of $R_0$ is high for all groups. The infecting time is higher in aged macaques than in young macaques.

## 3.2. Comparison of young and aged kinetics

In order to determine whether differences in parameter values between young and aged animals are significant, we compare the parameter distributions determined through bootstrapping. Parameter distributions for the ODE model are shown in figure 2 and $p$-values for the Mann–Whitney test are given in table 2. All parameters are statistically different in the nasal passages of young and aged animals, but some of these differences do not extend down into the throat. In the throat, we find that the infection rate and infectious cell death rate are statistically different. This is reflected in the parameter distributions that show large regions of overlap in the distributions of young and aged

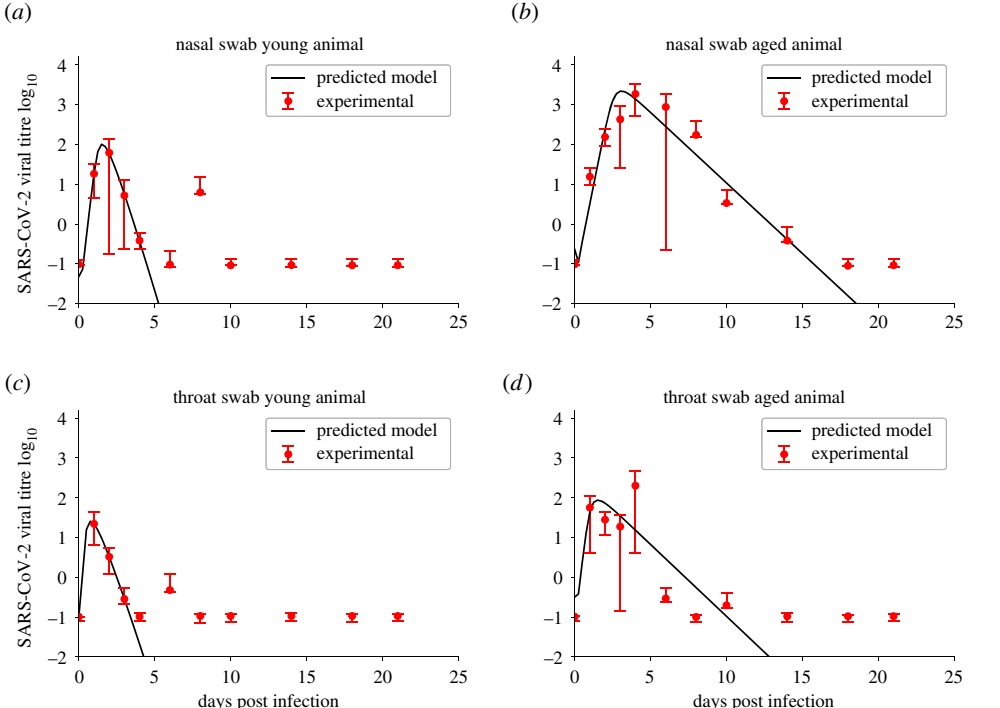

**Figure 1.** Model fits to experimental data from [38]. Experimental data are indicated by the red circles and viral kinetics (ODE) model best fits are given by solid black lines. Figures show the fits to data from nasal swabs of young (*a*) and aged (*b*) macaques and throat swabs of young (*c*) and aged (*d*) macaques.

**Table 1.** Best-fit parameter estimates for the ODE model.

| parameter | young nasal | aged nasal | young throat | aged throat |
|---|---|---|---|---|
| $\beta$ ((TCID$_{50}$ ml$^{-1}$)$^{-1}$ · (d)$^{-1}$) | 0.857 | $4.50 \times 10^{-3}$ | 3.94 | 0.286 |
| 95% confidence interval | 0.375–4.20 | $(1.19 - 17.0) \times 10^{-3}$ | 0.710–14.8 | 0.0177–2.26 |
| $p$ (TCID$_{50}$ ml$^{-1}$)$^{-1}$ · (d)$^{-1}$) | $4.02 \times 10^{3}$ | $4.05 \times 10^{4}$ | $7.40 \times 10^{2}$ | $2.32 \times 10^{3}$ |
| 95% confidence interval | $(2.62 - 12.9) \times 10^{3}$ | $(1.27 - 14.3) \times 10^{4}$ | $(0.240 - 22.5) \times 10^{3}$ | $(0.615 - 9.30) \times 10^{3}$ |
| $\delta$ (d) | 3.31 | 0.818 | 2.89 | 1.16 |
| 95% confidence interval | 3.19–30.4 | 0.598–1.14 | 2.51–61.4 | 0.768–2.45 |
| $V_0$ | $5.26 \times 10^{-6}$ | 0.238 | 0.102 | 0.102 |
| 95% confidence interval | $(0.137 - 6.47) \times 10^{-6}$ | $8.89 \times 10^{-3} - 1.14$ | 0.0240–0.128 | $8.82 \times 10^{-6} - 0.361$ |
| $R_0$ | 104 | 22.3 | 101 | 57.0 |
| 95% confidence interval | 26.0–108 | 12.2–47.0 | 11.8–525 | 15.4–481 |
| $t_{\text{inf}}$ (h) | 0.578 | 2.51 | 0.63 | 1.32 |
| 95% confidence interval | 0.345–0.697 | 1.76–3.21 | 0.125–1.62 | 0.502–2.28 |
| SSR | 3.23 | 1.89 | 0.460 | 2.35 |
| 95% confidence interval | $6.69 \times 10^{-6} - 4.44$ | 0.212–2.09 | $7.89 \times 10^{-12} - 1.27$ | 0.0177–3.26 |

animals in the throat for $R_0$ and $t_{\text{inf}}$ in particular, but clearly non-overlapping distributions for young and aged animals in the nasal passages.

## 3.3. Antiviral treatment

Since SARS-CoV-2 infections tend to be more severe in older adults [7,8], they are often treated early with antivirals in an effort to prevent serious illness [44]. We use our parameter estimates and the ODE model to investigate the effect of antiviral treatment that reduces production rate in young and aged animals.

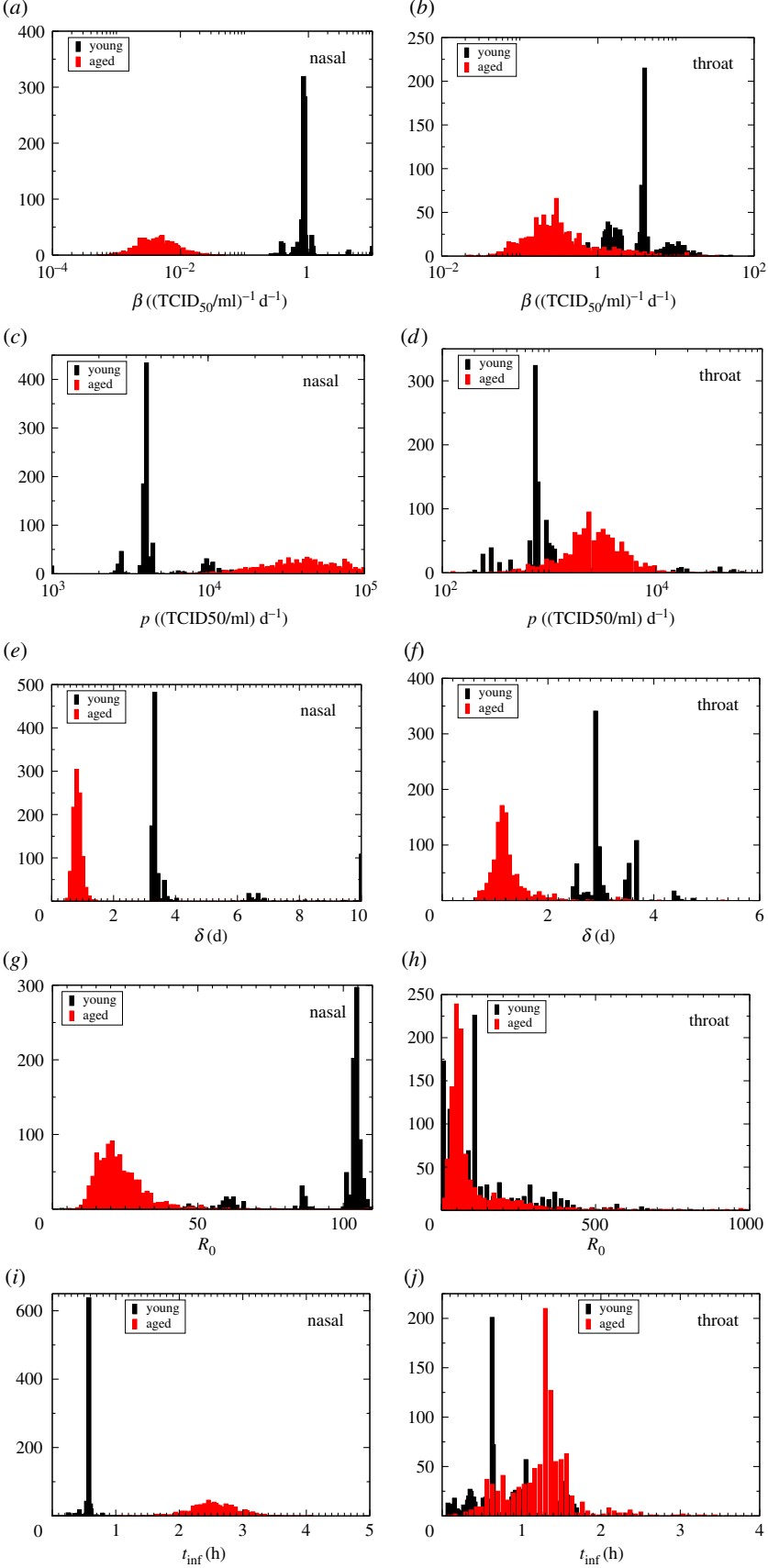

**Figure 2.** Parameter distributions for young and aged animals. ODE model parameter distributions estimated using bootstrapping are shown for nasal (*a,c,e,g,i*) and throat (*b,d,f,h,j*) swab data. Parameters shown are the infection rate (*a,b*), production rate (*c,d*), infectious cell death rate (*e,f*), basic reproduction number (*g,h*) and infecting time (*i,j*).

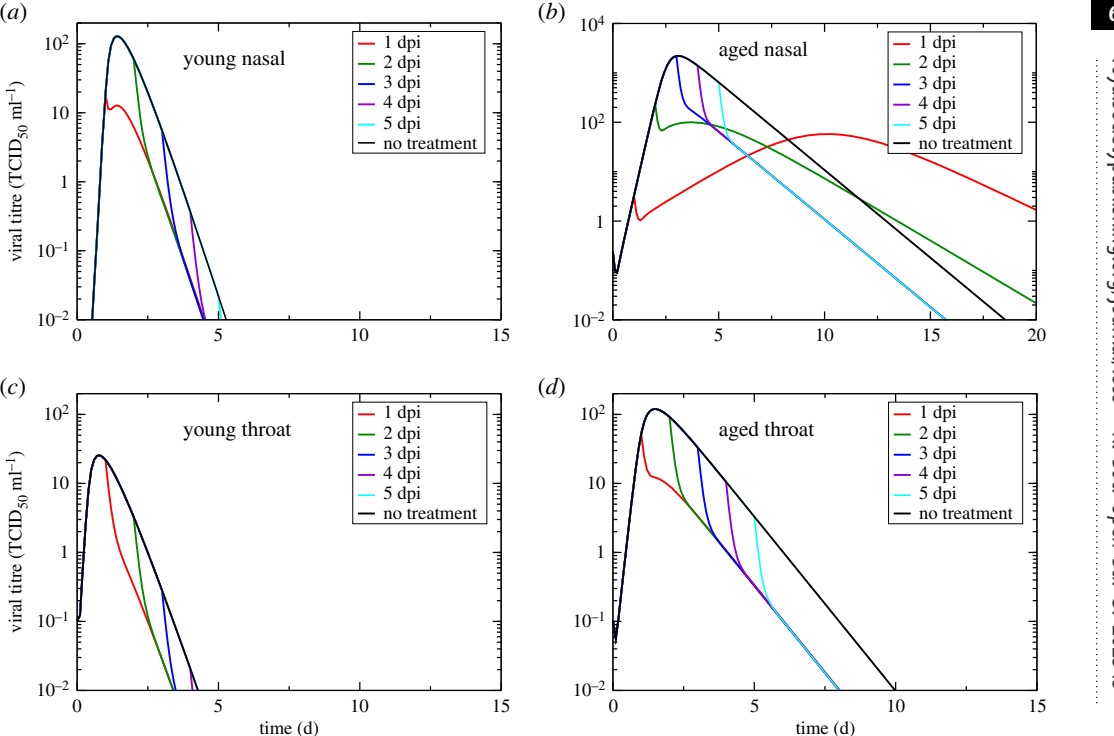

**Figure 3.** Model predictions of antiviral treatment. Figures show the predicted effect of an antiviral that reduces viral production in nasal passages of young animals (*a*), nasal passages of aged animals (*b*), throat of young animals (*c*) and throat of aged animals (*d*). The antiviral was assumed to reduce production by 90% and treatment was started at different days post-infection.

**Table 2.** Mann–Whitney *p*-values for comparing parameter distributions.

| parameter | nasal | throat |
|---|---|---|
| $\beta$ | $1.57 \times 10^{-4}$ | $2.06 \times 10^{-3}$ |
| $p$ | $1.93 \times 10^{-4}$ | 0.0412 |
| $\delta$ | $1.72 \times 10^{-4}$ | $4.63 \times 10^{-4}$ |
| $R_0$ | $1.03 \times 10^{-3}$ | 0.416 |
| $t_{inf}$ | $1.61 \times 10^{-4}$ | 0.0429 |

Examples of antivirals tested against SARS-CoV-2 that are modelled in this way include remdesivir [37,45–47], selinexor [46], lopinavir [40] and ritonavir [40]. We initially assume a drug effectiveness of 90% in reducing the production rate and examine the effect of treatment initiated at different times after infection onset. Untreated and treated viral titre curves are shown in figure 3.

In the throats of both young and aged animals as well as in the nasal passages of young animals, application of antivirals results in an immediate lowering of the viral load and a reduction of the infection duration. In the nasal passages of aged animals, antivirals result in an initial lowering of the viral load, but when the antiviral is applied before the viral titre peak, the viral load rises again after the initial reduction. This results in a longer-lasting infection with a delayed, lowered viral titre peak. If the dosage of antiviral is not sufficient to suppress production enough to suppress the infection, then reducing the production rate means that the amount of virus within the host increases, but slowly. If the viral load surpasses the threshold of detection, but is increasing slowly, the viral titre will remain above the threshold for a longer period of time. Since the lengthening of infection duration is combined with a lower viral titre peak, the overall effect on disease severity is not entirely clear.

We examined this more closely by measuring infection duration and peak viral load in nasal passages of young and aged animals over a range of drug efficacies and treatment initiation times. The results are shown in figure 4. Infection duration is defined here as the duration of time where viral titre is above the level of detection (0.1TCID$_{50}$ ml$^{-1}$). Note that in the figures, we present relative durations and relative

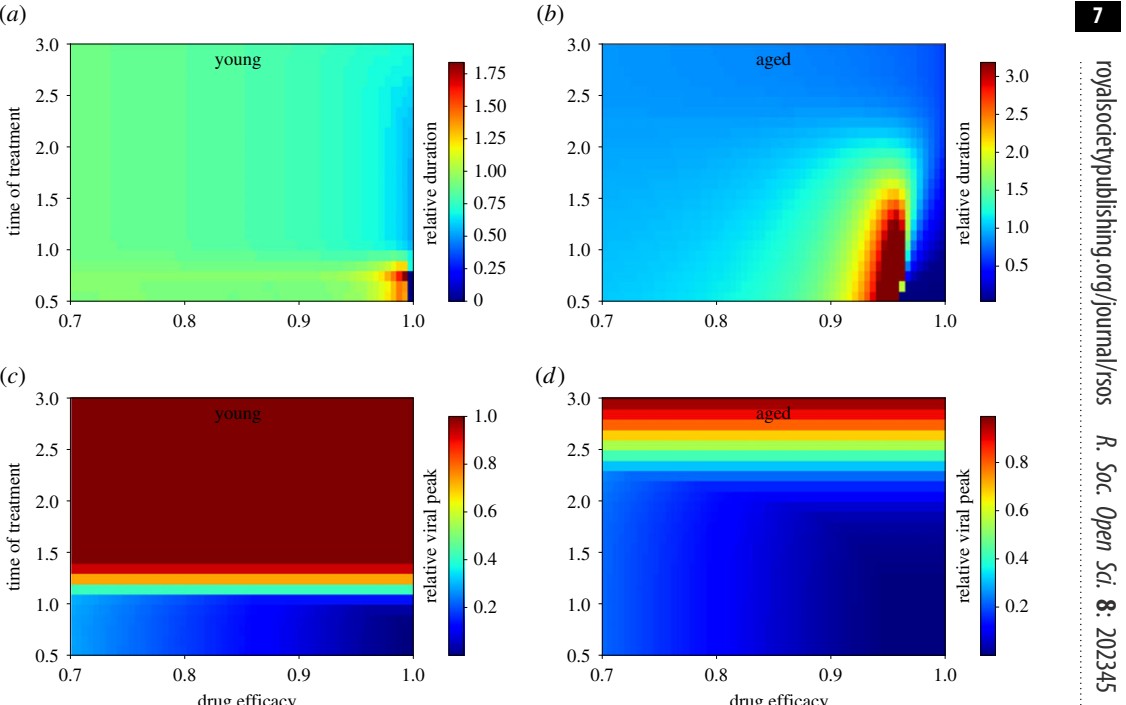

**Figure 4.** Effect of antiviral treatment on viral infections. (*a,b*) The infection duration in nasal passages of young (*a*) and aged (*b*) macaques. (*c,d*) The peak viral load in nasal passages of young (*c*) and aged (*d*) macaques. Infection durations and viral titre peaks are measured relative to those of an untreated infection.

viral peaks such that a value of 1 indicates that treatment does not change the measurement compared with an untreated infection. Values of relative duration greater than 1 indicate that drug treatment increases the infection duration and a number less than 1 indicates that the antiviral is having the intended effect. While most drug efficacies and treatment initiation times result in little change of the infection duration in the nasal passages of young animals, there is a small window at high drug efficacy and early treatment (before 1 dpi) where treatment leads to a lengthening of the infection. The treatment initiation times at which we would observe a lengthening of the infection in aged animals stretches to 2 dpi—a time period where treatment is more likely to be initiated. The viral titre peak is largely unaffected by drug efficacy since even drug efficacies of 70% are sufficient to cause immediate decay of the viral titre, but depends on the time at which treatment is initiated. Note that while the infection duration can be lengthened by antiviral treatment, viral titre peak will never be increased.

## 4. Discussion

We examined the differences in viral titre kinetics in the nose and throat of young and aged macaques by estimating best-fit parameters of an ODE model. We found statistically significant differences in all parameters describing the nasal viral kinetics of young and aged macaques, but noted fewer differences in the throat viral kinetics. Statistical comparisons were performed on parameter distributions estimated through bootstrapping, a technique used to determine variability in model parameter estimates, so do not reflect parameter variability due to animal-to-animal variation. There is some clinical evidence supporting the finding of differing viral dynamics at different locations in the respiratory tract. While not specifically age-related, a larger meta-analysis noted that severe illness was linked to longer viral shedding in the upper respiratory tract, but not in the lower respiratory tract [48], with a smaller study finding a similar result [49]. Other modelling studies are also finding differences in viral kinetics parameters between measurements taken at different locations in the respiratory tract in both macaques [41] and humans [50,51]. Biological and physiological conditions change as the virus moves down the respiratory tract. For example, the distribution of cell types and their associated receptors, particularly the ACE2 receptor used by SARS-CoV-2, changes along the respiratory tract [52–54], with higher prevalence of ACE2 in the nasal passages [52]. Since we observe

changes in all model parameters, it is not clear from this study what biological changes might be leading to the differences in viral dynamics in the young and old.

We show that the dynamical differences between infection in young and aged animals has an impact on the effectiveness of antiviral treatment. In particular, antiviral treatment initiated before the viral titre peak can lead to prolonged infections, albeit with a lower viral burden. Since the viral titre peak is delayed in older adults as compared with young adults, this provides both an increased opportunity to administer antiviral treatment that might be beneficial, but there is also an increased opportunity to enter the region where antiviral treatment leads to prolonged infection. Since older adults are more susceptible to serious illness and death, they are likely to be treated early with antivirals [44], potentially hitting this window and prolonging the length of time over which they are shedding virus. The importance of timing of initiation of antiviral treatment relative to the viral titre peak has been noted in previous modelling studies [55–57], even for SARS-CoV-2 [40], but is not clear whether longer shedding of a lower amount of virus is beneficial from a clinical perspective [58].

Two recent studies have estimated viral kinetic model parameters for SARS-CoV-2 infections in non-human primates [37,41], so we can compare our parameter estimates with other studies. This comparison can only be performed with quantities that do not include viral units, since viral measurements are not standardized and can vary from one experiment to the next [33]. Infectious cell death rates are similar— 0.8–3.3 d$^{-1}$ here compared with 1.5–3 d$^{-1}$ in [37]—and also compare well with a mean infectious cell death rate of 1.88 d$^{-1}$ found in cynomolgus macaques by Gonçalves *et al.* [41]. Values of the basic reproduction number found in [41] were lower (mean of 6 and 4) than in this study (22–104) and in [37] (12–85). Finally, [41] also noted very short infecting times (approx. 0.1 h) compared with the infecting times found here (0.6–2.5 h) and in [37] (1–2 h). It should also be noted that parameter values found in non-human primates do not necessarily translate to humans, where modelling studies are finding lower values of the basic reproduction number [40,50,59] and longer infecting times [40,59].

One of the limitations of this model is the lack of an explicit immune response. While mathematical models that include an immune response are being investigated for SARS-CoV-2 [50,59–61], there is still a paucity of clinical data describing the time course of immune responses, making it difficult to properly parametrize these models. This is particularly problematic for studying the underlying causes of differences between young and elderly patients since many of the proposed reasons for the differing infection severity are related to the immune response [21–26]. Mathematical models that include an immune response have been used to study the effect of ageing in influenza [62,63], finding connections among varied immune responses that differ among young and aged mice. Such insight, however, is only possible with detailed time course data for the immune responses being examined. It is also important to note that many severe symptoms are not directly related to viral load or to cell damage by virus, but rather are caused by the immune response [64–66]. While some models have attempted to relate disease severity and mortality to model variables [67–69], the mechanistic link is not clear and this is still an open problem.

The analysis here is also limited by the data. We used averaged viral titre measurements from several animals as representative time courses for each group. A recent study has indicated that making measurements on averaged data does not always lead to the same result as making measurements on individual time courses and averaging the measurements [70]. The datasets also consist of a limited number of points, which limits our ability to identify some parameters. To remedy this, we fixed some parameters ($k$ and $c$) that are known to not be independently identifiable for this model [71]. Despite model limitations, this study has found differences in viral dynamics of SARS-CoV-2 between young and aged animals and has shown that these differences should be considered when considering antiviral treatment.

Ethics. This manuscript uses previously published data from animal studies; no animal studies were performed specifically for this study. As such, no ethics approval was required.

Data accessibility. Data and relevant code for this research work are stored in GitHub: https://github.com/hdobrovo/COVID_aging.git and have been archived within the Zenodo repository: https://doi.org/10.5281/zenodo.5137827.

Authors' contributions. Conceptualization, H.M.D.; methodology, T.R. and H.M.D.; software, T.R.; validation, T.R. and H.M.D.; formal analysis, T.R.; writing—original draft preparation, H.M.D.; writing—review and editing, T.R. and H.M.D.; supervision, H.M.D.; project administration, H.M.D. All authors have read and agreed to the published version of the manuscript.

Competing interests. The authors have no competing interests to declare.

Funding. No funding has been received for this article.

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
