## [Peer Review File · Royal Society Open Science]

Review History

RSOS-202345.R0 (Original submission)

Review form: Reviewer 1

Is the manuscript scientifically sound in its present form?

Yes

Are the interpretations and conclusions justified by the results?

No

Is the language acceptable?

Yes

Do you have any ethical concerns with this paper?

No

Have you any concerns about statistical analyses in this paper?

Yes

Recommendation?

Major revision is needed (please make suggestions in comments)

Comments to the Author(s)

Please see the review report (Appendix A).

Decision letter (RSOS-202345.R0)

Dear Dr Dobrovolny

The Editors assigned to your paper RSOS-202345 "Mathematical modeling finds differences in viral kinetics in nasal passages, but not in the throat, of young and aged SARS-CoV-2-infected macaques" have now received comments from reviewers and would like you to revise the paper in accordance with the reviewer comments and any comments from the Editors. Please note this decision does not guarantee eventual acceptance.

Please submit your revised manuscript and required files (see below) no later than 21 days from today's (ie 14-Jun-2021) date. Note: the ScholarOne system will 'lock' if submission of the revision is attempted 21 or more days after the deadline. If you do not think you will be able to meet this deadline please contact the editorial office immediately.

Kind regards,

Royal Society Open Science Editorial Office
Royal Society Open Science
openscience@royalsociety.org

on behalf of Dr Shigui Ruan (Associate Editor) and Glenn Webb (Subject Editor)

Reviewer comments to Author:

Reviewer: 1

Comments to the Author(s)

Please see the review report (pdf).

===PREPARING YOUR MANUSCRIPT===

===PREPARING YOUR REVISION IN SCHOLARONE===

Please ensure that you include a summary of your paper at Step 2 'Type, Title, & Abstract'. This should be no more than 100 words to explain to a non-scientific audience the key findings of your

research. This will be included in a weekly highlights email circulated by the Royal Society press office to national UK, international, and scientific news outlets to promote your work.

Author's Response to Decision Letter for (RSOS-202345.R0)

See Appendix B.

RSOS-202345.R1 (Revision)

Review form: Reviewer 1

Is the manuscript scientifically sound in its present form?

Yes

Are the interpretations and conclusions justified by the results?

Yes

Is the language acceptable?

Yes

Do you have any ethical concerns with this paper?

No

Have you any concerns about statistical analyses in this paper?

Yes

Recommendation?

Accept as is

Comments to the Author(s)

I appreciate the authors for working on the revision in detail. The authors have adequately addressed all the comments I had. I recommend to accept this revised manuscript for publication in Royal Society Open Science.

Decision letter (RSOS-202345.R1)

Dear Dr Dobrovolny,

It is a pleasure to accept your manuscript entitled "Estimation of viral kinetics model parameters in young and aged SARS-CoV-2 infected macaques" in its current form for publication in Royal Society Open Science. The comments of the reviewer(s) who reviewed your manuscript are included at the foot of this letter.

COVID-19 rapid publication process:

We are taking steps to expedite the publication of research relevant to the pandemic. If you wish, you can opt to have your paper published as soon as it is ready, rather than waiting for it to be published the scheduled Wednesday.

This means your paper will not be included in the weekly media round-up which the Society sends to journalists ahead of publication. However, it will still appear in the COVID-19 Publishing Collection which journalists will be directed to each week (<https://royalsocietypublishing.org/topic/special-collections/novel-coronavirus-outbreak>).

If you wish to have your paper considered for immediate publication, or to discuss further, please notify openscience_proofs@royalsociety.org and press@royalsociety.org when you respond to this email.

on behalf of Dr Shigui Ruan (Associate Editor) and Glenn Webb (Subject Editor)
openscience@royalsociety.org

Reviewer comments to Author:

Reviewer: 1

Comments to the Author(s)

I appreciate the authors for working on the revision in detail. The authors have adequately addressed all the comments I had. I recommend to accept this revised manuscript for publication in Royal Society Open Science.

Appendix A

Review Report: Manuscript “**Mathematical modeling finds differences in viral kinetics in nasal passages, but not in the throat, of young and aged SARS-CoV-2-infected macaques**” by *Rodriguez and Dobrovlny*.

In this manuscript, authors used a basic viral dynamic model to study viral kinetics in nasal passages and throats of young and aged SARS-CoV-2 infected macaques. Based on the parameters estimated from the data fitting to the model, the authors claim that the difference in viral kinetics occur in nasal passages, but not in the throat of young and aged animals. Furthermore, they simulated the model to observe viral dynamics for different treatment timing. While the study of viral dynamics of SARS-CoV-2 within a host, including young and aged hosts, is very important, the analysis performed in this study is limited to make assertive claim that the authors have made. I have some concerns which need to be addressed before accepting for publication. My major concerns are:

- Only one model is used ignoring all other possible mechanism. Justify why this model is the most suitable: can basic model without having eclipse phase describe the data? Is it okay to absolutely ignore the immune responses?
- There are number of animals in both groups in the experiment, but only one data set for each group is considered for data fitting. Is it average?. At least, the standard errors should be indicated if the distinction to be made between the groups.
- The claim based on one data set in each group is hard to justify, especially when there are extremely limited viral data points above the detection limit. For example, you have only 3-5 data points above the limit of detection in young animals. It is in fact not clear or not mentioned how many data points were actually considered for fitting. The data points considered, including those below the limit of detection, might highly affect the parameter estimates.
- The nature of data clearly shows that the data in nasal and throat of young animals are almost the same (1-2 days to reach peak 4-5 days to reach below the limit of detection) and also the data in nasal and throat of aged animals are almost the same (4-5 days to reach peak 10-15 days to reach below the limit of detection). Therefore, I am not fully convinced that your claim on having the different dynamics in nasal passages, but the same dynamics in throat in young and aged animals. The different or the same values of parameters obtained between animals could be due to fitting error. For example, you observed δ in throat of young animal to be lower making it similar to aged animal. It could have been because the solution curve was pulled towards the later data points, causing poor fitting; note that almost all the data points were missed in your fitting in throat of young animals (Fig 1). It may be possible to improve. It is known that post peak decay is higher for higher δ . My main point is: it is very hard to claim this without further thorough analysis.
- The parameters estimated is based on one fixed set of values for k and c , which may vary from animal to animal (especially they may also be different for those two groups). It is important to do analysis on: does the result remain the same for different k and c values? What if they are also estimated as it is quite possible that they are different for young and aged animals?
- The statistical significance observed is misleading because these were not obtained based on number of subjects. It is based on data generated through bootstrapping from the models with distinct parameters. So, they are not independent animals data but the data generated from the same parameters with variations. Such claims are misleading.
- Is it enough to measure effectiveness of antiviral therapy, just based on infection duration? What about the level of viral load (peak, etc.)?

- What could be the biological reason to have the intended effect (longer infection duration) of treatment?
- Please explain how you obtained the expression of t_{inf} . It may not be obvious for many readers.
- Why $T(0) = 1$ (page 15)? Was it scaled?
- Manuscript needs to be read carefully as it is written with error in number of places. For example:
 - Where is the dashed blue lines in Fig 1? It was pointed out in the figure caption.
 - It is mentioned young and aged “mice” in Figure 4. Mice?

Appendix B

Response to Reviewers for: Estimation of viral kinetics model parameters in young and aged SARS-CoV-2 infected macaques

Thalia Rodriguez and Hana M. Dobrovolny

July 23, 2021

Below, we respond to the reviewers' comments and provide a clear description and exact location for any change made to the manuscript to address their comments. We would like to thank the reviewers for their comments.

Reviewer 1

In this manuscript, authors used a basic viral dynamic model to study viral kinetics in nasal passages and throats of young and aged SARS-CoV-2 infected macaques. Based on the parameters estimated from the data fitting to the model, the authors claim that the difference in viral kinetics occur in nasal passages, but not in the throat of young and aged animals. Furthermore, they simulated the model to observe viral dynamics for different treatment timing. While the study of viral dynamics of SARS-CoV-2 within a host, including young and aged hosts, is very important, the analysis performed in this study is limited to make assertive claim that the authors have made. I have some concerns which need to be addressed before accepting for publication. My major concerns are:

1.1 Only one model is used ignoring all other possible mechanism. Justify why this model is the most suitable: can basic model without having eclipse phase describe the data? Is it okay to absolutely ignore the immune responses?

Note that a model without an eclipse phase will not have fewer free parameters since we have fixed the duration of the eclipse phase. That said, it is possible to fit the data with a simpler model without an eclipse phase, but we get best fits with higher SSRs for most of the data sets (see Table below), so we presented the results for the model with an eclipse phase in the manuscript.

Data Set	Eclipse SSR	No Eclipse SSR
Young Nasal	3.23	4.04
Aged Nasal	1.89	1.80
Young Throat	0.46	0.469
Aged Throat	2.35	2.70

We have added this information to the supplementary material.

The issue with more complex models is that they have more parameters that need to be estimated. As the reviewer later points out, the number of data points in each data set is small. While we can fit more complex models to the data and probably get lower SSRs, some of these parameters will not be identifiable, making comparisons of parameter estimates between data sets difficult. While the immune response is not explicitly included, the effect of the immune response is implicitly incorporated in the parameter value estimates since the immune response will change the average death rate of cells (via killing of infected cells by cytotoxic T lymphocytes) or could change the infection or production rates (via interferon).

1.2 There are number of animals in both groups in the experiment, but only one data set for each group is considered for data fitting. Is it average? At least, the standard errors should be indicated if the distinction to be made between the groups.

We do not have access to viral time courses in individual animals, only the averaged time courses presented in [14]. We have added the standard errors to Fig. 1.

1.3 The claim based on one data set in each group is hard to justify, especially when there are extremely limited viral data points above the detection limit. For example, you have only 3-5 data points above the limit of detection in young animals. It is in fact not clear or not mentioned how many data points were actually considered for fitting. The data points considered, including those below the limit of detection, might highly affect the parameter estimates.

It is common practice to combine multiple measurements into an average time course, for in vitro [11, 12], animal [6], and human challenge studies [2, 10], and make comparisons between the average time courses for the groups. A typical example of this is in determining treatment efficacy of antiviral compounds when researchers assess whether the antiviral shortens the duration of an infection. So while it is one data set in each group, they are not data sets based on measurements from single individuals, but are measurements averaged over several individuals.

The data set with the smallest number of points above threshold is the time course of virus in the throat of young animals with 4 data points above threshold. The low number of data points above threshold is part of the reason we fixed some of the parameters — by fixing some data points that are not independently identifiable [16], we hopefully improve the identifiability of the parameters that are left. As mentioned in the Methods section, “for

experimental data at the detection threshold, model estimates only contribute to the SSR if they are above the threshold.” So while all points are included in the fitting, the points at threshold near the end of the infection are taken into account while those at later days do not really contribute to the SSR.

1.4 The nature of data clearly shows that the data in nasal and throat of young animals are almost the same (1-2 days to reach peak 4-5 days to reach below the limit of detection) and also the data in nasal and throat of aged animals are almost the same (4-5 days to reach peak 10-15 days to reach below the limit of detection). Therefore, I am not fully convinced that your claim on having the different dynamics in nasal passages, but the same dynamics in throat in young and aged animals. The different or the same values of parameters obtained between animals could be due to fitting error. For example, you observed δ in throat of young animal to be lower making it similar to aged animal. It could have been because the solution curve was pulled towards the later data points, causing poor fitting; note that almost all the data points were missed in your fitting in throat of young animals (Fig 1). It may be possible to improve. It is known that post peak decay is higher for higher δ . My main point is: it is very hard to claim this without further thorough analysis.

We disagree with the assessment of the reviewer that the viral time courses are similar in the nose and throat of both young and aged animals. We agree that the time courses in the nose and throat of young animals have similar time of peak and time to resolution. In aged animals, however, the viral titer time course appears to peak sooner and resolve sooner in the throat of aged animals — this appears to be what the model fits are capturing. That said, we have gone back and re-fit all the data sets using several different algorithms available as options in `scipy.minimize`. This has allowed us to find slightly better fits for all data sets, with the largest decrease in SSR occurring in the fit to the young throat data where the best fit curve now goes through all the earlier time points, but misses the measurement at $t = 6$ d. This has led to additional statistically significant differences — with R_0 in the nose now also being statistically different between young and old, as well as β and δ being different in the throat. Consequently, we have changed the title and modified the manuscript accordingly.

As an additional assessment of whether we have the best fit, we examined the likelihood profiles for all the estimated parameters (shown below and included in the supplemental material).

In all cases, the profile has a local minimum, with our best fit estimate falling at the minimum.

1.5 The parameters estimated is based on one fixed set of values for k and c , which may vary from animal to animal (especially they may also be different for those two groups). It is important to do analysis on: does the result remain the same for different k and c values? What if they are also estimated as it is quite possible that they are different for young and aged animals?

We agree with the reviewer that it is possible that k and c are different for young and aged animals. However, with just viral titer data, only one of δ , k or c is identifiable for the target cell limited model used here [16]. In fact, when all three of these parameters are allowed to vary, the parameter estimation often assigns them the same value when using this model to fit viral titer data only [1, 3, 13]. Since the data is limited, as the reviewer points out, and these parameters are not independently identifiable, we chose to fix them, as has been done in other recent COVID modeling studies [7, 8]. To assess the effect of the chosen values of k and c , we examine the likelihood profiles for these variables. The likelihood profiles for k are shown below.

We see that even though k was not free to vary during the minimization process, the fixed values are near the minimal SSR. The likelihood profiles for c are shown below.

These are also near the minimal SSR values.

As an additional check, we re-fit the data assuming values of $k = 1/d$ or $k = 5/d$ and $c = 5/d$ or $c = 20/d$. As shown in the table below, in most cases, the SSR is lowest for the fixed values of $k = 3/d$ and $c = 10/d$; we have highlighted fits with lower SSR in bold.

New fixed value	Data set	SSR
$k = 1$ d	Aged Nasal	1.59
	Young Nasal	4.24
	Aged Throat	2.43
	Young Throat	1.23
$k = 5$ d	Aged Nasal	1.96
	Young Nasal	3.23
	Aged Throat	2.41
	Young Throat	0.460
$c = 5$ d	Aged Nasal	1.74
	Young Nasal	3.23
	Aged Throat	2.33
	Young Throat	0.460
$c = 20$ d	Aged Nasal	2.04
	Young Nasal	3.22
	Aged Throat	2.35
	Young Throat	0.469

While there are a some lower SSRs, there is no particular value of k or c that improves fits for all the data sets. This assessment of the choice of k and c is now also included in the supplementary material.

1.6 The statistical significance observed is misleading because these were not obtained based on number of subjects. It is based on data generated through bootstrapping from the models with distinct parameters. So, they are not independent animals data but the data generated from the same parameters with variations. Such claims are misleading.

This type of statistical comparison has been used in other studies [11, 12, 15] to compare parameter estimates for different viral infection conditions. The basic idea is the same as

when statistically comparing measurements from different animal cohorts. In the case of multiple animals, the error in the parameter values is estimated using animal-to-animal variation. We do not have measurements from multiple animals, so we need a different method for estimating the error in our parameter values — bootstrapping is a common technique for estimating error in parameter values determined via model fitting [4, 5]. For this reason, we do not believe the results are misleading. However, we have gone through the manuscript and added text to clarify that the statistical analysis is not based on animal to animal variation and have added the following to the discussion section “Statistical comparisons were performed on parameter distributions estimated through bootstrapping, a technique used to determine variability in model parameter estimates, so do not reflect parameter variability due to animal-to-animal variation.”

1.7 Is it enough to measure effectiveness of antiviral therapy, just based on infection duration? What about the level of viral load(peak,etc.)?

We have added the plots for viral peak to Fig. 4.

1.8 What could be the biological reason to have the intended effect (longer infection duration) of treatment?

If the dosage of antiviral is not sufficient to suppress production enough to suppress the infection, then reducing the production rate means that the amount of virus within the host increases, but slowly. If the viral load surpasses the threshold of detection, but is increasing slowly, the viral titer will remain above the threshold for a longer period of time. This lengthening of infection duration is combined with a lower viral titer peak (as now shown in the manuscript), so the overall effect on disease severity is not entirely clear. We have included these comments in the manuscript.

1.9 Please explain how you obtained the expression of t_{inf} . It may not be obvious for many readers.

A derivation of t_{inf} is found in [9]. We now point the reader to that reference in the text.

1.10 Why $T(0) = 1$ (page 15)? Was it scaled?

We are using units of relative cell number for the cells. We have noted this in the text.

1.11 Manuscript needs to be read carefully as it is written with error in number of places. For example: Where is the dashed blue lines in Fig 1? It was pointed out in the figure caption. It is mentioned young and aged “mice” in Figure 4. Mice?

We have corrected both of these.

References

- [1] P. Baccam, C. Beauchemin, C. A. Macken, F. G. Hayden, and A. S. Perelson. Kinetics of influenza A virus infection in humans. *J. Virol.*, 80(15):7590–7599, August 2006. doi: 10.1128/JVI.01623-05.
- [2] B. Bagga, C. W. Woods, T. H. Veldman, A. Gilbert, A. Mann, G. Balaratnam, R. Lambkin-Williams, J. S. Oxford, M. T. McClain, T. Wilkinson, B. P. Nicholson, G. S. Ginsburg, and J. P. DeVincenzo. Comparing influenza and RSV viral disease dynamics in experimentally infected adults predicts clinical effectiveness of RSV antivirals. *Antivir. Ther.*, 18:785–791, 2013. doi: 10.3851/IMP2629.
- [3] C. A. Beauchemin, J. J. McSharry, G. L. Drusano, J. T. Nguyen, G. T. Went, R. M. Ribeiro, and A. S. Perelson. Modeling amantadine treatment of influenza A virus in vitro. *J. Theor. Biol.*, 254:439–451, 21 September 2008. doi: 10.1016/j.jtbi.2008.05.031.
- [4] A. C. Davison and D. V. Hinkley. *Bootstrap methods and their application*. Cambridge ; New York, NY, USA : Cambridge University Press, 1997. ISBN 0521573912 (hb). Reprinted with corrections 1998, 1999, 2000, 2003.
- [5] B. Efron and R. Tibshirani. *An introduction to the bootstrap*. New York : Chapman and Hall, 1993. ISBN 0412042312. Includes indexes.
- [6] T. Enkirch, S. Sauber, D. E. Anderson, E. S. Gan, D. Kenanov, S. Maurer-Stroh, and V. von Messling. Identification and in vivo efficacy assessment of approved orally bioavailable human host protein-targeting drugs with broad anti-influenza A activity. *Frontiers Immunol.*, 10:1097, June 5 2019. doi: 10.3389/fimmu.2019.01097.
- [7] A. Gonçalves, J. Bertrand, R. Ke, E. Comets, X. de Lamballerie, D. Malvy, A. Pizzorno, O. Terrier, M. R. Calatrava, F. Mentré, P. Smith, A. S. Perelson, and J. Guedj. Timing of antiviral treatment initiation is critical to reduce SARS-CoV-2 viral load. *CPT Pharmacomet. Sys. Pharmacol.*, 9(9):509–514, September 2020. doi: 10.1002/psp4.12543.
- [8] A. Gonçalves, P. Maisonnasse, F. Donati, M. Albert, S. Behillil, V. Contreras, T. Naninck, R. Marlin, C. Solas, A. Pizzorno, J. Lemaitre, N. Kahlaoui, O. Terrier, R. H. T. Fang, V. Enouf, N. Dereuddre-Bosquet, A. Brisbarre, F. Touret, C. Chapon, B. Hoen, B. Lina, M. Rosa-Calatrava, X. de Lamballerie, F. Mentré, R. L. Grand, S. van der Werf, and J. Guedj. Viral dynamic modeling of SARS-CoV-2 in non-human primates. *PLoS Comput. Biol.*, 17(3):e1008785, March 17 2021. doi: 10.1371/journal.pcbi.1008785.
- [9] G. González-Parra, F. De Ridder, D. Huntjens, D. Roymans, G. Ispas, and H. M. Dobrovlny. A comparison of RSV and influenza in vitro kinetic parameters reveals differences in infecting time. *Plos One*, 13(2):e0192645, 8 February 2018. doi: 10.1371/journal.pone.0192645.

- [10] F. G. Hayden, J. J. Treanor, R. F. Betts, M. Lobo, J. D. Esinhart, and E. K. Hussey. Safety and efficacy of the neuraminidase inhibitor GG167 in experimental human influenza. *JAMA*, 275(4):295–299, January 1996.
- [11] E. Paradis, L. Pinilla, B. Holder, Y. Abed, G. Boivin, and C. Beauchemin. Impact of the H275Y and I223V mutations in the neuraminidase of the 2009 pandemic influenza virus in vitro and evaluating experimental reproducibility. *PLoS ONE*, 10(5):e0126115, May 2015. doi: 10.1371/journal.pone.0126115.
- [12] L. T. Pinilla, B. P. Holder, Y. Abed, G. Boivin, and C. A. A. Beauchemin. The H275Y neuraminidase mutation of the pandemic A/H1N1 influenza virus lengthens the eclipse phase and reduces viral output of infected cells, potentially compromising fitness in ferrets. *J. Virol.*, 86(19):10651–10660, October 2012. doi: 10.1128/JVI.07244-11.
- [13] L. Pinky and H. M. Dobrovoly. Coinfections of the respiratory tract: Viral competition for resources. *PLoS ONE*, 11(5):e0155589, May 19 2016. doi: 10.1371/journal.pone.0155589.
- [14] B. Rockx, T. Kuiken, S. Herfst, T. Bestebroer, M. M. Lamers, B. B. O. Munnink, D. de Meulder, G. van Amerongen, J. van den Brand, N. M. Okba, D. Schipper, P. van Run, L. Leijten, R. Sikkema, E. Verschoor, B. Verstrepen, W. Bogers, J. Langermans, C. Drosten, M. F. van Vliissingen, R. Fouchier, R. de Swart, M. Koopmans, and B. L. Haagmans. Comparative pathogenesis of COVID-19, MERS, and SARS in a nonhuman primate model. *Science*, 368(6494):1012–1015, May 29 2020. doi: 10.1126/science.abb7314.
- [15] T. Rodriguez and H. M. Dobrovoly. Quantifying the effect of trypsin and elastase on in vitro SARS-CoV infections. *Virus Res.*, 299:198423, May 2021. doi: 10.1016/j.virusres.2021.198423.
- [16] A. M. Smith, F. R. Adler, and A. S. Perelson. An accurate two-phase approximate solution to an acute viral infection model. *J. Math. Biol.*, 60(5):711–726, May 2010. doi: 10.1007/s00285-009-0281-8.